# Prevalence of second mesiobuccal canal in maxillary molars of Iranian population: A systematic review with meta-analysis

Seyed Mohsen Hasheminia[1], Saber Khazaei[2], Pedram Iranmanesh [1], Jafar Kolahi[1], Bita Rasteh[3], Masoumeh Behdarvandi [4]*

1 Department of Endodontics, Dental Research Center, Dental Research Institute, School of Dentistry, Isfahan University of Medical Sciences, Isfahan, Iran, 2 Department of Endodontics, School of Dentistry, Kermanshah University of Medical Sciences, Kermanshah, Iran, 3 Ahvaz Jundishapur University of Medical Sciences, Ahvaz, Iran, 4 Department of Endodontics, School of Dentistry, Kashan University of Medical Sciences, Kashan, Iran

* masbhdad@gmail.com

## Abstract

One of the main reasons for the failure of root canal treatment in maxillary molars is missing the second mesiobuccal (MB2) canal. This systematic review and meta-analysis aimed to estimate the prevalence of the MB2 canal in maxillary molars in the Iranian population.: Cross-sectional studies on the prevalence of MB2 canal in maxillary molars in the Iranian population were searched in PubMed, Scopus, Magiran, SID, IranMedex, and Google Scholar databases until 31 May 2024. The JBI Critical Appraisal Tool for prevalence studies was utilized for quality assessment. The prevalence of the MB2 canal was estimated with a 95% confidence coefficient (95%CI) using STATA18 software and the inverse variance method, separately for the first and second molars (CRD42024542862).In the present study, the prevalence of the MB2 canal in the maxillary first molar was 60% (95%CI, 53_67; $I^2 = 97.14\%$) and 33% (95%CI, 25_42; $I^2 = 97.42\%$) in the maxillary second molar. The highest prevalence rate of the MB2 canal in the maxillary first and second molars related to the microscope with 78% (95%CI, 54_101; $I^2 = 97.96\%$) and 61% (95%CI, 20_102%; $I^2 = 97.43\%$), respectively. The lowest prevalence rate of the MB2 canal in the maxillary first and second molars was related to the periapical radiograph with a rate of 15% (95%CI, −10_40; $I^2 = 91.17\%$) and 4% (95% CI, −8_17; $I^2 = 40.52\%$), respectively. The prevalence of the MB2 canal in the Iranian population in the first and second molars was 60% and 33%, respectively. This prevalence rate depended on the assessment methods.

## Introduction

Missing an additional canal, was identified as one of the most common failure reasons for root canal treatment [1]. Several methods have been used to identify

**Data availability statement:** All relevant data are within the paper and its Supporting Information files. Also, the data underlying the results presented in the study are available from Figshare via https://doi.org/10.6084/m9.figshare.28070612.

**Funding:** SM.H received the grant by Isfahan University of Medical Sciences (#3400125). The funders had no role in study design, data collection and analysis, decision to publish, or preparation of the manuscript.

**Competing interests:** The authors have declared that no competing interests exist.

the additional canal, particularly the second mesiobuccal (MB2) canal of maxillary molars. These methods include sectioning, staining, and clearing, periapical (PA) radiography, cone beam computed tomography (CBCT), micro-CT, electron scanning [2], direct vision [3] and magnification with a loupe or microscope [4,5].

Although most studies have shown the presence of the MB2, there is no conclusion about its prevalence among diverse populations [6]. For instance, the prevalence was 63.6% in Thaiwan, 89.5% in South African, 59.5%in Polish, 61.9% in American, and ranged between 44.0 and 88.5% in Brazillian populations [7]. This controversies among different reports might be related to the method of assessment and the race, age [6,8], or gender [8] of the study population.

The morphology of the canal and the root varies widely in different races and even in different people of the same race. It is necessary for the clinician to have sufficient knowledge about the configuration of the root canal. Considering that, a meta-analysis study has not been conducted in the Iranian population, the present systematic review and meta-analysis aimed to estimate the prevalence of the MB2 canal in the maxillary molars in the Iranian population.

## Materials and methods

### Study design and eligibility criteria

Study protocol was approved by the Iranian National Committee for Ethics in Biomedical Research (IR.MUI.RESEARCH.REC.1400.082), and the protocol was registered at International Prospective Register of Systematic Reviews (PROSPERO) (CRD42024542862) [9]. The present systematic review was reported based on the PRISMA checklists [10] (S1 and S2 Tables). The research question, based on PEOS framework was: What is the prevalence (O) of the MB2 canal (E) in the maxillary molars of the Iranian population (P) in cross-sectional studies (S)? The inclusion criteria were cross-sectional studies that investigated the prevalence of the MB2 canal in maxillary teeth in the Iranian population.

### Search strategy

Records published until 31 May 2024 were searched in PubMed, Scopus, Magiran, IranMedex, and Scientific Information Database (SID), and 100 first hit of Google Scholar databases. The search included articles published in Persian and English, with no time restriction.

The search strategy was (((mesiobuccal OR mesi* OR MB OR MB2) OR (Canal AND (morphology* OR anatomy* OR configuration*))) AND (Prevalence OR frequency OR incidence) AND (Upper OR Maxilla*) AND Molar* AND Iran*) in the title, abstract, and keywords/subject. Additionally, the references list of the included articles was searched.

### Extracting the data

After obtaining the records, duplicates were removed. The screening was done by examining the title and abstract of the records, and in the next step, the whole text was screened to obtain the records, based on the eligibility criteria. The name of the

first author, year of publication, study location, sample size, evaluation method, and prevalence of the MB2 canal for each record were extracted. All stage was done by two researchers separately (S.KH and M.B) and any disagreement was solved through discussion with the third researcher (P.I).

## Quality assessment

The Joanna Briggs Institute (JBI) Critical Appraisal Tool for prevalence studies (S3 Table) was used to assess the quality of individual studies [11]. Two criteria on coverage bias (Q5) and response rate (Q9) were not considered. The studies assessment was performed by two evaluators independently (J.K and M.B.), and any discrepancies were debated. The studies with JBI score of equal or higher than 50 were included.

## Statistical methods

The prevalence of the MB2 canal was analysed by STATA 18 software (StataCorp, College Station, TX) using the restricted maximum likelihood method. Heterogeneity was evaluated by the $I^2$ statistic and the $I^2$ statistic values higher than 50% were considered as high heterogeneity. Galbraith plots were applied to figure out the outliers' studies. Sensitivity tests were carried out by removing each record. Publication bias was investigated by Egger's regression and Begg's adjusted rank correlation test. Visual inspection for publication bias was complemented with nonparametric trim-and-fill analysis to estimate the number of potential missing records. A $P < 0.05$ was considered significant.

# Results

## Description of included records

A total of 586 records were obtained. After removing duplicates, the titles and abstracts of 250 records were screened (S4 Table). After excluding 210 studies, 40 records were reviewed by reading the whole text, of which 19 were excluded (S5 Table). In addition, 8 records were retrieval by grey literature. All included studies received a JBI Score higher than 50; therefore, no study was excluded in this regard (S6 and S7 Tables). Finally, 29 records were included in a systematic review and meta-analysis (Fig 1). Out of 29 recodes, 27 [2,4,5,12–35] and 17 records [4,12,15,17–22,24,27,29,32,33,35–37] were considered first and second maxillary molar, respectively.

## Characteristics of included studies

A total of 7510 maxillary molars had been studied, of which 4390 and 3120 were maxillary first and second molars, respectively. Furthermore, out of 27 studies for the maxillary first molar, 23 studies used one method, including 15 studies were done by CBCT [2,13–21,23–27], five by staining and clearing [28–32], one by sectioning [34], one by PA radiography [33] and one microscope [35]. Four studies used multiple methods: Hasheminia et al. [12] used sectioning, and staining and clearing; Ghorbanzade et al. [4] used direct vision, microscope, and loupe; Zand et al. [22] used CBCT and PA radiography; and Khademi et al. [5] used CBCT, microscope, and micro-CT. Finally, 33 datasets were included for qualitative and quantitative syntheses (Table 1).

Out of 17 studies for the maxillary second molar, 15 studies used one method: nine were done by CBCT [15,17–21,24,27,36], four by staining and clearing [12,29,32,37], one microspore [35] and one study by PA radiography [33]. Two studies used multiple methods including Ghorbanzadeh et al. [4] which used direct vision, microscope, and loupe and Zand et al. [22] used PA radiography and CBCT. Finally, 20 datasets were included for qualitative and quantitative syntheses (Table 2).

## Meta-analysis outcome

For the maxillary first molar, the prevalence of the MB2 canal was 60% (95%CI, 53_68; $I^2 = 97.14\%$). According to the method of investigation, the highest prevalence was related to the microscope method, with a prevalence of 78% (95%CI,

PRISMA 2020 flow diagram for new systematic reviews which included searches of databases, registers and other sources

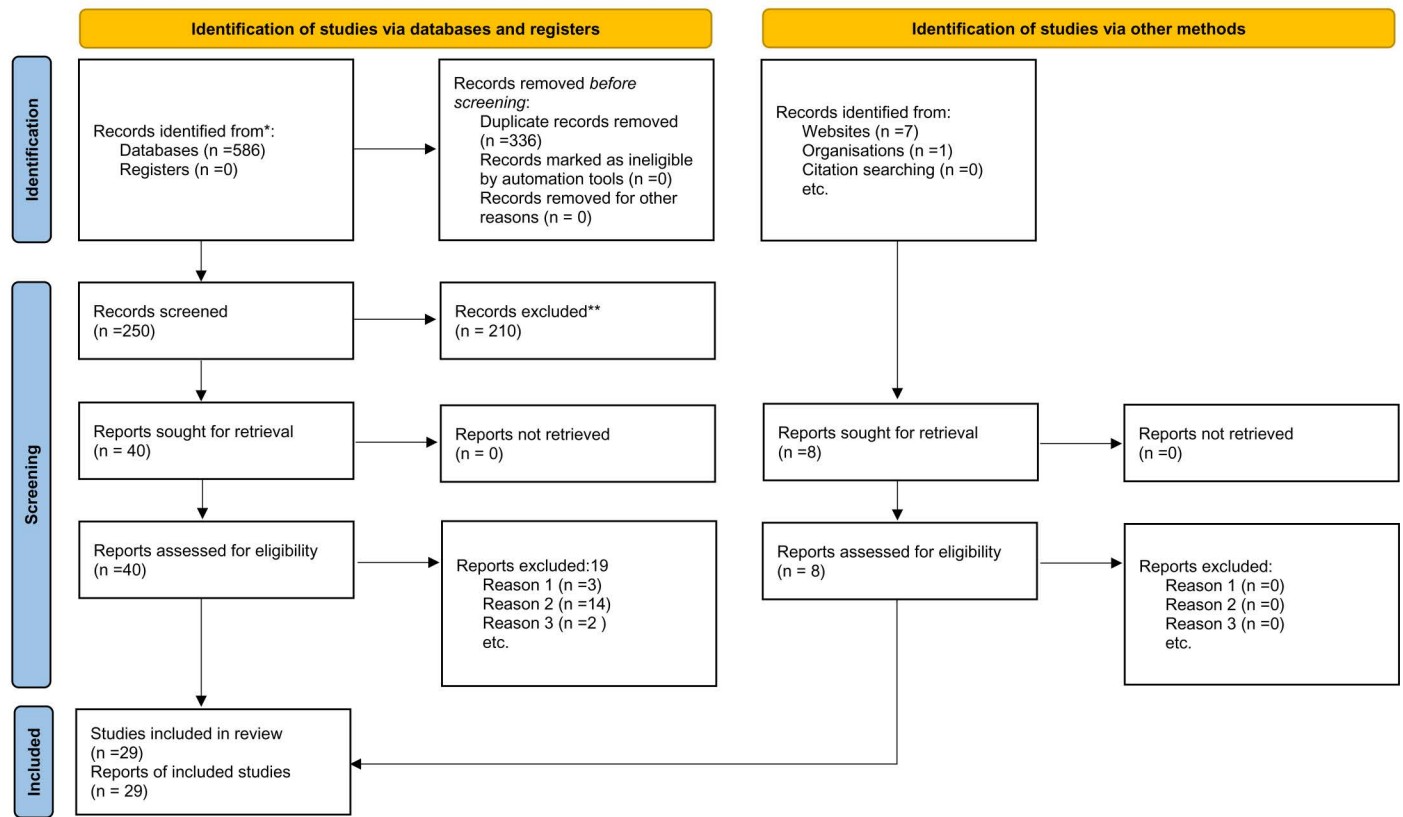

*Consider, if feasible to do so, reporting the number of records identified from each database or register searched (rather than the total number across all databases/registers).
**If automation tools were used, indicate how many records were excluded by a human and how many were excluded by automation tools.

Source: Page MJ, et al. BMJ 2021;372:n71. doi: 10.1136/bmj.n71.

This work is licensed under CC BY 4.0. To view a copy of this license, visit https://creativecommons.org/licenses/by/4.0/

**Fig 1.  PRISMA 2020 flow diagram.**

54_101; $I^2 = 97.96\%$), followed by the sectioning and micro-CT methods with 77% (95%CI, 70_83; $I^2 = 0\%$) and 76% (95%CI, 66_87; $I^2 = 0\%$), respectively. In contrast, the lowest prevalence of MB2 canal was related to the PA radiography method with a prevalence of 15% (95%CI, −10_40; $I^2 = 91.17\%$) (Fig 2). The Galbraith plot (Fig 3) of the maxillary first molar, most studies were clustered around the regression line and fell within the 95% confidence interval, suggesting that the overall effect size is consistent across studies. However, a few studies were located toward the edges of the confidence interval, which could suggest slight heterogeneity or variation in effect sizes across studiesThese plots showed that two studies including Naseri et al. [29] and Zand (B) et al. [22] were outlier. However, the sensitivity test showed the overall percentage was robust (Fig 4).

For the maxillary second molar, the prevalence of the MB2 canal was 33% (95%CI, 25_42; $I^2 = 97.42\%$). According to the method of investigation, the highest prevalence of MB2 canal was related to the microscope method with a prevalence of 61% (95%CI, 20_102%; $I^2 = 97.43\%$), followed by the loupe with a prevalence of 56% (95%CI, 40_71; $I^2 = 0\%$). In contrast, the lowest prevalence was related to the PA radiography method with a prevalence of 4% (95% CI, −8_17; $I^2 = 40.52\%$) (Fig 2). The overall pattern of Galbraith plot for maxillary second molare was similar to the maxillary first molar, but there seems to be a wider spread in effect sizes, with some studies closer to or outside the boundaries of the

**Table 1. Data extraction of the maxillary first molar.**

| Author, year | Location | Prevalence (%) | Sample size | Type of study | Quality assessment score | Methods |
|---|---|---|---|---|---|---|
| Sharifi, 2023 | Kermanshah | 88 | 213 | Cross-sectional | 100 | CBCT |
| Namdar, 2023 | Mazandaran | 71.3 | 348 | Cross-sectional | 100 | |
| Khademi (A), 2022 | Isfahan | 70.4 | 71 | Cross-sectional | 100 | |
| Dibaji, 2022 | Tehran | 50.1 | 311 | Cross-sectional | 100 | |
| Karkehabadi, 2022 | Hamedan | 70 | 193 | Cross-sectional | 86 | |
| Esmaeilian, 2021 | Isfahan | 67.5 | 200 | Cross-sectional | 100 | |
| Nikkerdar 2020 | Kermanshah | 51 | 250 | Cross-sectional | 86 | |
| Tafakhori, 2018 | Rafsanjan | 63 | 41 | Cross-sectional | 86 | |
| Khosravifard, 2018 | Rasht | 45 | 145 | Cross-sectional | 100 | |
| Ghoncheh, 2017 | Tehran | 46 | 345 | Cross-sectional | 86 | |
| Zand (A), 2017 | Tabriz | 55 | 156 | Cross-sectional | 100 | |
| Ghaznavi, 2017 | Urmia | 41 | 167 | Cross-sectional | 100 | |
| Khademi, 2016 | Isfahan | 70 | 389 | Cross-sectional | 100 | |
| Naseri, 2016 | Tehran | 87 | 149 | Cross-sectional | 100 | |
| Faramarzi, 2015 | Hamedan | 69 | 156 | Cross-sectional | 100 | |
| Ezoddini Ardakani, 2014 | Yazd | 60 | 30 | Cross-sectional | 86 | |
| Rouhani, 2014 | Mashhad, Tehran, Tabriz, Bandar Abbas, Isfahan | 54 | 125 | Cross-sectional | 86 | |
| Rezaeian, 2018 | Rafsanjan | 61 | 80 | Cross-sectional | 86 | Staining clearing |
| Naseri, 2015 | Tehran | 11 | 35 | Cross-sectional | 86 | |
| Adel, 2009 | Qazvin | 75 | 114 | Cross-sectional | 72 | |
| Shahi, 2007 | Tabriz | 58 | 137 | Cross-sectional | 100 | |
| Hasheminia (A), 2005 | Isfahan | 61 | 80 | Cross-sectional | 100 | |
| Sadeghi, 2004 | Rafsanjan | 76 | 50 | Cross-sectional | 100 | |
| Zand (B), 2017 | Tabriz | 3 | 156 | Cross-sectional | 100 | Periapical radiography |
| Safi, 2000 | Shiraz | 29 | 42 | Cross-sectional | 72 | |
| Ashofteh Yazdi, 2005 | Tehran | 75 | 105 | Cross-sectional | 86 | Sectioning |
| Hasheminia (B), 2005 | Isfahan | 79 | 80 | Cross-sectional | 100 | |
| Ghorbanzadeh (A), 2009 | Tehran | 27 | 45 | Cross-sectional | 86 | Direct vision |
| Parirokh, 2023 | Kerman | 72.3 | 333 | Cross-sectional | 100 | |
| Khademi (C), 2022 | Isfahan | 59.2 | 71 | Cross-sectional | 100 | Microscope |
| Ghorbanzadeh (B), 2009 | Tehran | 99 | 45 | Cross-sectional | 86 | |
| Ghorbanzadeh (C), 2009 | Tehran | 64 | 45 | Cross-sectional | 86 | Loupe |
| Khademi (B), 2022 | Isfahan | 76.1 | 71 | Cross-sectional | 100 | Micro-CT |

95% confidence interval. This may indicate greater heterogeneity among the studies in this subset or grouping. According to the this plot, Ghorbanzadeh (B) et al. [4] was an outlier study (Fig 3). However, the sensitivity test indicated the overall percentage was robust (Fig 4).

## Publication bias

No publication bias was observed based on Egger's regression and Begg tests for both groups (Table 3). Yet, the non-parametric trim-and-fill analysis showed seven potentially missed studies for the maxillary first molar (Fig 5). All included studies showed lower prevalence in comparison with pooled prevalence. No missed study were found for the maxillary second molar and the funnel plot was symmetrical (Fig 5).

**Table 2. Data extraction of the maxillary second molar.**

| Author, year | Location | Prevalence (%) | Sample size | Type of study | Quality assessment score | Methods |
|---|---|---|---|---|---|---|
| Namdar, 2023 | Mazandaran | 37.1 | 375 | Cross-sectional | 100 | CBCT |
| Karkehabadi, 2022 | Hamedan | 35.5 | 193 | Cross-sectional | 86 | |
| Esmaeilian, 2021 | Isfahan | 23.5 | 200 | Cross-sectional | 100 | |
| Nikkerdar, 2020 | Kermanshah | 34 | 250 | Cross-sectional | 86 | |
| Naseri, 2018 | Tehran | 68 | 157 | Cross-sectional | 100 | |
| Khosravifard., 2018 | Rasht | 19 | 135 | Cross-sectional | 100 | |
| Ghoncheh, 2017 | Tehran | 14 | 423 | Cross-sectional | 86 | |
| Zand (A), 2017 | Tabriz | 27 | 156 | Cross-sectional | 100 | |
| Khademi, 2016 | Isfahan | 43 | 460 | Cross-sectional | 100 | |
| Rouhani, 2014 | Mashhad, Tehran, Tabriz, Bandar Abbas, Isfahan | 19 | 125 | Cross-sectional | 86 | |
| Naseri, 2015 | Tehran | 40 | 35 | Cross-sectional | 86 | Staining clearing |
| Zarei, 2009 | Mashhad | 52 | 103 | Cross-sectional | 100 | |
| Hasheminia (A), 2005 | Isfahan | 24 | 80 | Cross-sectional | 100 | |
| Sadeghi, 2004 | Rafsanjan | 24 | 50 | Cross-sectional | 100 | |
| Zand (B), 2017 | Tabriz | 1 | 156 | Cross-sectional | 100 | Periapical radiography |
| Safi, 2000 | Shiraz | 17 | 12 | Cross-sectional | 72 | |
| Ghorbanzadeh (A), 2009 | Tehran | 16 | 45 | Cross-sectional | 86 | Direct vision |
| Parirokh, 2023 | Kerman | 40.2 | 333 | Cross-sectional | 100 | |
| Ghorbanzadeh (B), 2009 | Tehran | 82 | 45 | Cross-sectional | 86 | Microscope |
| Ghorbanzadeh (C), 2009 | Tehran | 56 | 45 | Cross-sectional | 86 | Loupe |

## Discussion

In the present study, the pooled prevalence of the MB2 canal in the maxillary first molar (60%) was higher than the second molar (33%), which was similar to other studies [38]. In various populations, a range of 25% to 93.5% and 30% to 50% have been reported for the first and second molars, respectively (S8 Table). In a systematic review and meta-analysis which assessed global prevalence of MB2 using CBCT [8], pooled prevalence of MB2 canal was higher in maxillary first molar (69.6%; 64.5%−74.8%) than in second molars (39.0%; 31.1%−46.9%). In Indian population a pooled prevalence of 64.76% for MB2 in permanent maxillary first molars was reported using CBCT [3]. The difference in the reported prevalence of MB2 among countries may be due to variations in evaluation methods and racial groups [6,8].

Natural selection influences the size and structure of teeth based on environmental and adaptive needs. In populations where large teeth are advantageous (e.g., for better survival or dietary reasons, as seen in African populations), this trait is maintained and becomes dominant. Conversely, in environments where this selective pressure is absent, smaller teeth tend to evolve due to factors like dietary changes, reduced need for mechanical processing, or cultural practices, reflecting an adaptation to less physically demanding conditions [8,39]. Based on the evolutionary evidence and the influence of external anatomy on the internal morphology of teeth, the different prevalence of the MB2 canal in maxillary molars is under the expected geographical location [40], and the present outcomes cannot be generalized to other populations.

In the present study, the highest prevalence of the MB2 canal was reported with microscope, sectioning, micro-CT and loupe methods. Using a microscope with 10X and 16X magnification showed the highest prevalence for both molars. A

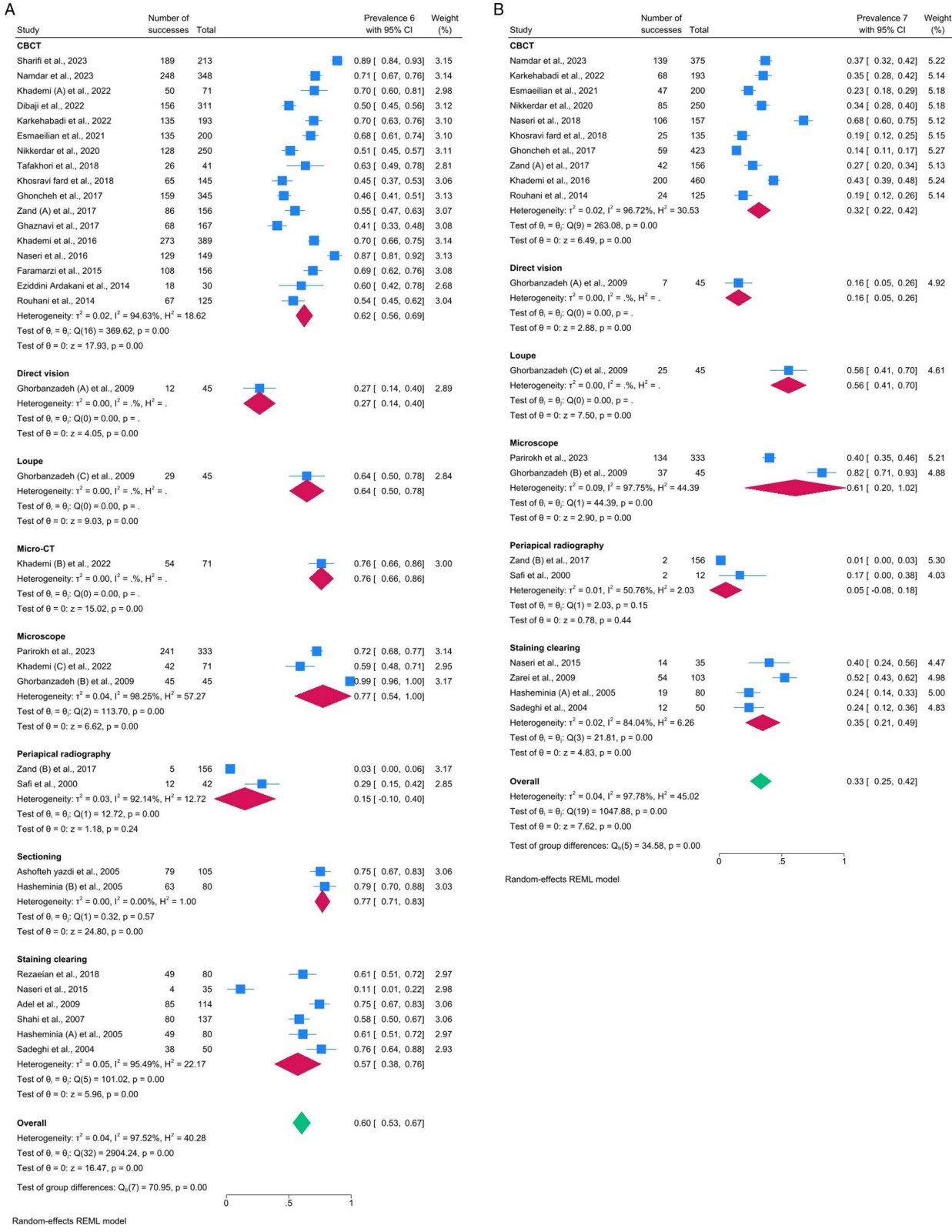

**Fig 2. Forest plots for maxillary (a) first and (b) second molar.**

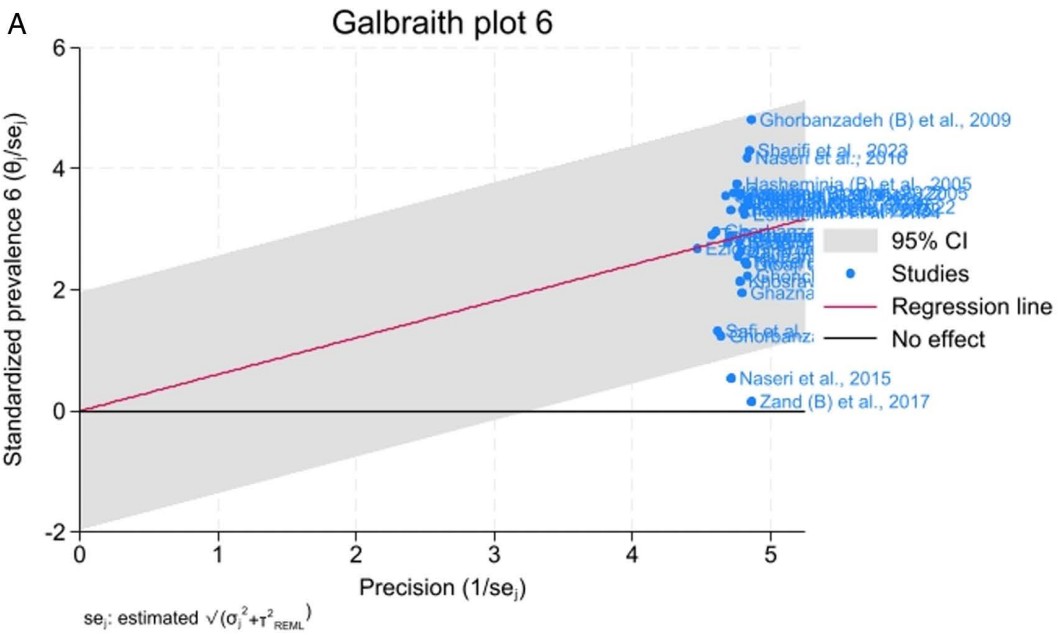

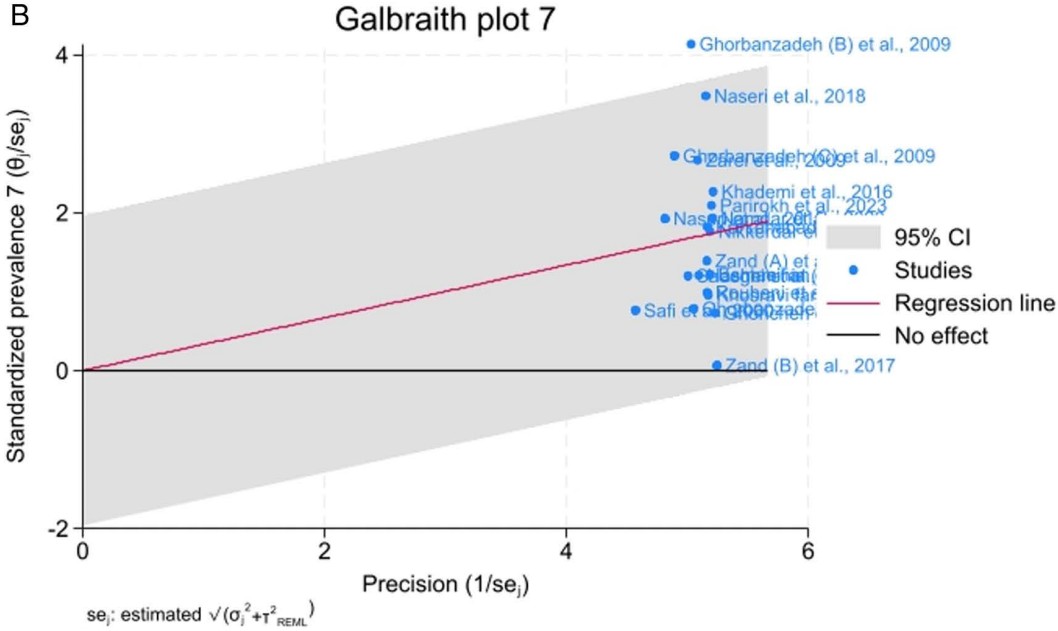

**Fig 3. Galbraith plots for maxillary (a) first and (b) second molar.**

high prevalence was also reported in an in vitro study where the MB2 canal was investigated under a microscope [41]. The troughing procedure may be effective for finding the MB2 canal when using the microscope [42]. Using a loupe with 3.5X magnification and a headlight, a study showed a high prevalence of MB2 for maxillary first and second molars [4]. Hence, using magnification and the troughing procedure can be a practical approach to identify and address the presence of an MB2 canal in molars.

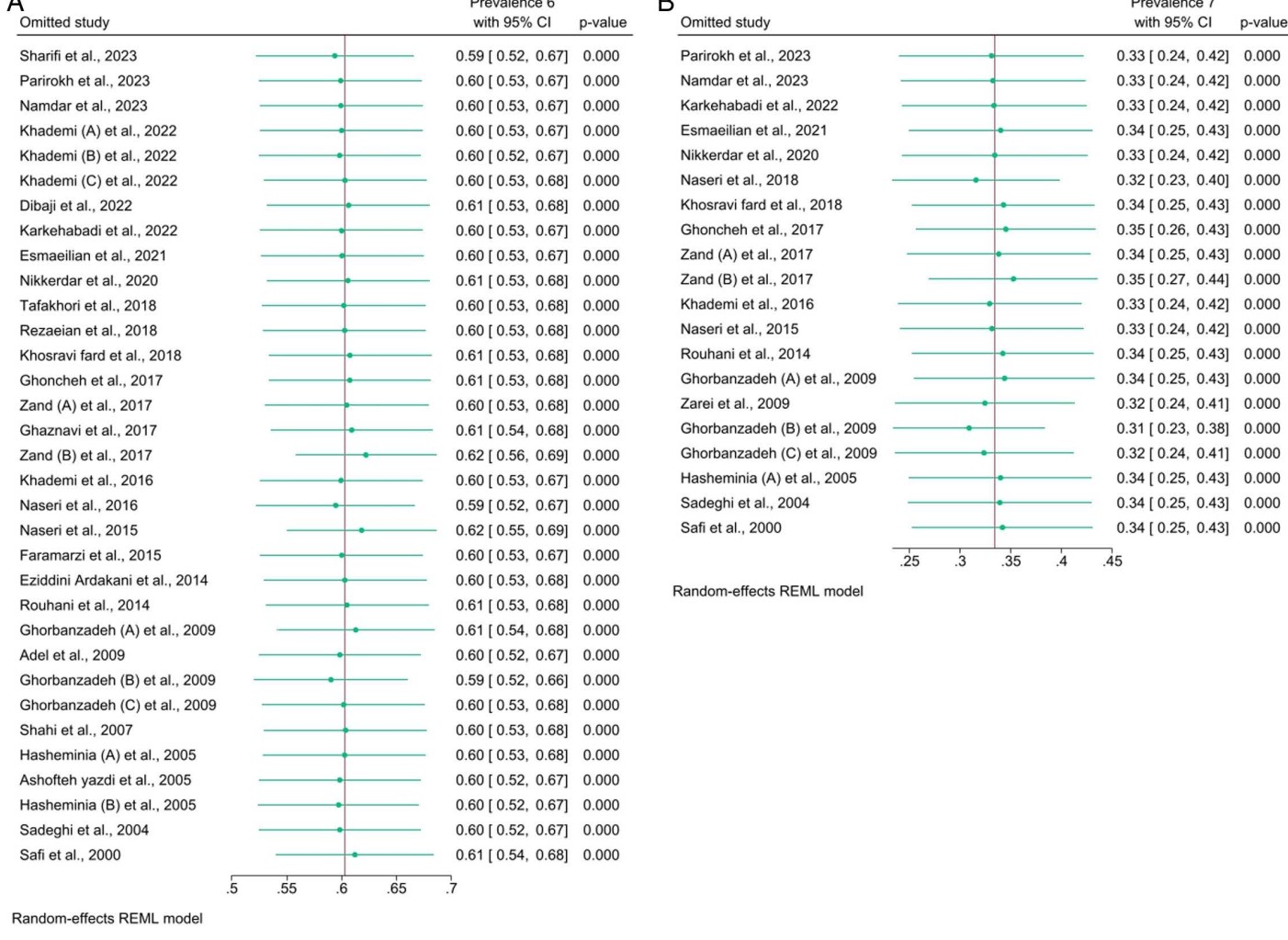

**Fig 4. Sensitivity tests for maxillary (a) first and (b) second molar.**

**Table 3. Egger's regression and Begg tests.**

| Group | Egger's test | P-value | Kendall's Tau | P-value |
|---|---|---|---|---|
| First maxillary molar | −1.96 | 0.32 | −105.00 | 0.10 |
| Second maxillary molar | 1.67 | 0.35 | 0.00 | 0.99< |

The prevalence rate of the MB2 canal in the first maxillary molars in the sectioning method was reported to be about 77%. This method was used in no study in the maxillary second molar. This technique is invasive and requires tooth extraction and some variables such as age, sex, and time of tooth eruption would not report in this method [23]; thus, this method is not recommended anymore.

Micro-CT, an effective non-destructive tool for evaluating the intricate three-dimensional structure of the root canal system, was used in a study [5]. Compared to CBCT, it offers pointedly higher spatial resolution, making it the gold standard

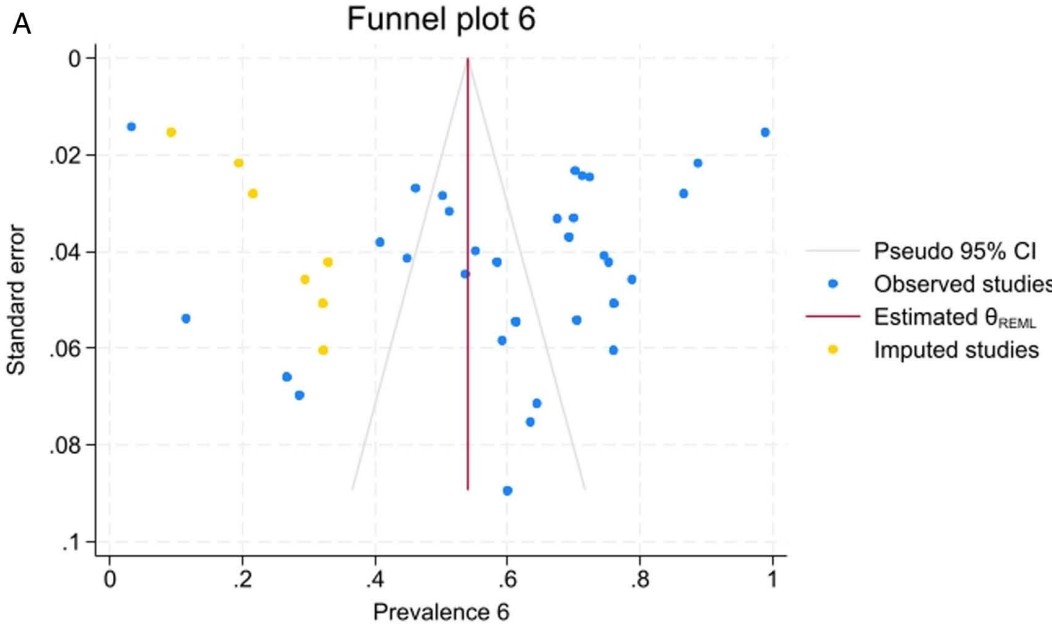

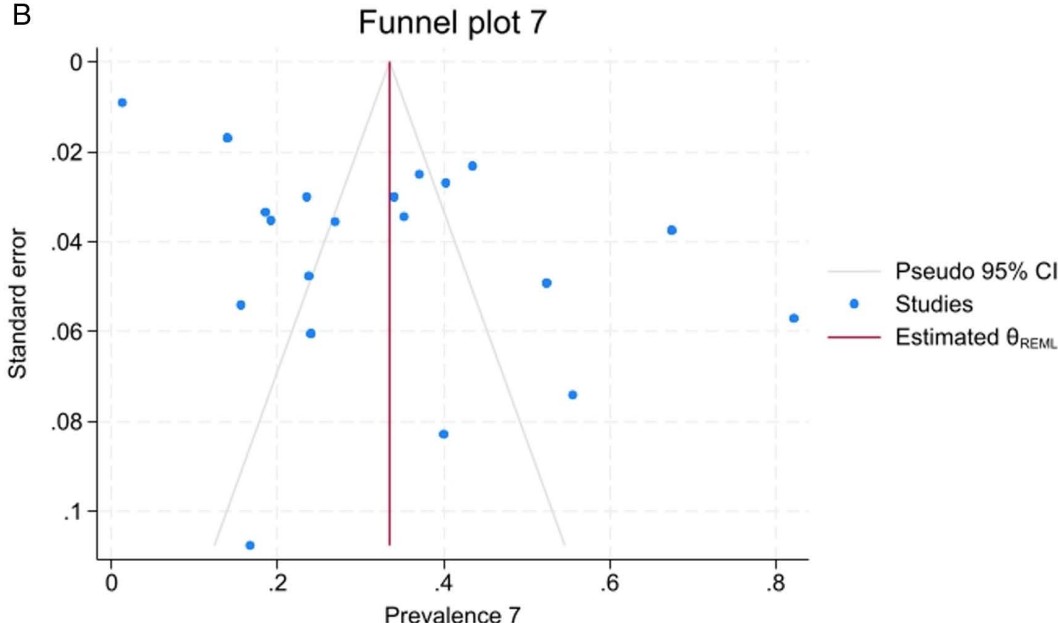

**Fig 5. Funnel plot with nonparametric trim-and-fill analysis for maxillary first (a) and second (b) molar.** For maxillary first molar, seven potentially missed records were identified and the percentage of second mesiobuccal canals when considering those decreased to 54.1 (95% confidence interval 46.7 to 61.5). For the maxillary second molar, no potentially missed records were noted.

for comprehensive assessment of root canal morphology [5]. In others hand, many studies used CBCT method considering high resolution, significant reduction of patient radiation dose, fast performance, and low cost compared to micro-CT [21]. In the CBCT method, the prevalence of the MB2 canal in maxillary molars was 62% and 32% in the maxillary first molar and second molar, respectively. In addition to race, differences in the voxel size (from 75 to 320 μm) could account

for the differences in reported percentages (S9 and S10 Tables). However, the optimal voxel size for the detection of MB2 was not determined.

Other methods that were used to evaluate MB2 were staining and clearing. The prevalence rates of the MB2 canal in this method were estimated to be 57% and 35% for maxillary first and second molars, respectively. A wide range has been mentioned in different studies, which can be attributed to gender, racial, and geographical differences. Although this technique is difficult and time-consuming, it has advantages, such as maintaining the original morphology of the root canal, three-dimensional representation of the root canal, and lateral and accessory canals [28].

The lowest prevalence rate of MB2 has been reported in direct vision (27% and 16% for maxillary first and second molars, respectively) and PA radiography (15% and 4% for maxillary first and second molars, respectively) methods. The troughing procedure, along with direct vision, increases the prevalence of the MB2 canal [39]. In addition, a PA radiograph is a two-dimensional image of a three-dimensional object, which causes image distortion and superimposition, so some details might lost [21]. Considering the disadvantages of these methods, investigating the prevalence of the MB2 canal by other methods is prudent.

Among the factors that may affect the prevalence of MB2 are gender and age. In a meta-analysis [8], the prevalence of MB2 canal in the maxillary first molar was higher in men than in women. The differential effects of X and Y-chromosomes on growth cause larger dimensions of permanent and deciduous teeth in men than in women [43]. This explains the larger molars in men and possibly the increase in the number of root canals. In the present meta-analysis, a few studies investigated the effect of gender on the prevalence of the MB2 canal; thus, it was impossible to evaluate this issue. Some studies concluded that due to calcification in the older adults, fewer canals were detected in the mesiobuccal root [44]. Contrarily, no relationship or a lower prevalence of the MB2 canal in the older adults was reported by other studies [45]. That may related to continuous deposition of dentin [45]. In the present study, analysis could not be performed in this regard due to the small number of studies reporting ages.

### Limitation

This systematic review and meta-analysis have several limitations. First, the variation in the methodologies, including differences in imaging techniques and sample sizes, could introduce heterogeneity and information bias [46,47]. Due to the small number of studies reporting prevalence of MB2 according to ages and gender, the analysis could not be conducted in this regard. Additionally, the generalizability of the findings may be limited due to the focus on Iranian populations, which may not fully reflect the global prevalence of the MB2 canal in maxillary molars. Finally, there was a discrepancy between Egger's regression and Begg's tests and nonparametric trim-and-fill analysis that might be related to their different assumptions. Trim-and-fill accounts for potential missing studies based on the observed asymmetry, even it is not statistically significant in routine tests [48].

### Future implication

This study was the first meta-analysis investigating the MB2 prevalence in the Iranian population, which was reported separately by different evaluation methods. Future studies are suggested by accurately reporting the age, gender, and geographic location of the teeth, as well as using modern techniques, such as the microscope, loupe and particularly CBCT in vivo which might best describes all variables including side (right and left), age, and gender while being less expensive and less invasive than micro-CT

### Conclusion

It was concluded that the prevalence of the MB2 canal was higher in the first molar (60%) than in the second molar (33%) of the maxilla, and the prevalence of the MB2 canal depended on the assessment method. It seems that the differences in the reported prevalence of the outcome under study are due to the difference in the technique used to evaluate the

patients, which can produce heterogeneous results. Knowing that each technique depends on how much sensitivity and specificity it has for the diagnosis of the disease, it is possible in future research to convert the prevalence values and to standardize the prevalence associated with each technique to obtain a result with less heterogeneity. It is recommended that this approach be used for future meta-analyses.

## Supporting information

**S1 Table.  PRISMA 2020 checklist.**
(DOCX)

**S2 Table.  PRISMA 2020 for abstracts checklist.**
(DOCX)

**S3 Table.  Joanna Briggs Institute (JBI) critical appraisal checklist for studies reporting prevalence data.**
(DOCX)

**S4 Table.  A list of 250 retrieval records from databases after removing duplicate entries.**
(XLSX)

**S5 Table.  Excluded studies at the full text assessment phase with reason.**
(DOCX)

**S6 Table.  Quality assessment of included studies of maxillary first molars according to Joanna Briggs Institute (JBI) critical appraisal checklist for studies reporting prevalence data.**
(DOCX)

**S7 Table.  Quality assessment of included studies of maxillary second molars according to Joanna Briggs Institute (JBI) critical appraisal checklist for studies reporting prevalence data.**
(DOCX)

**S8 Table.  Overall MB2 root canal prevalence in maxillary first and second molars in the different counties.**
(DOCX)

**S9 Table.  Overall MB2 root canal prevalence in maxillary first molars according to voxel size.**
(DOCX)

**S10 Table.  Overall MB2 root canal prevalence in maxillary second molars according to voxel size.**
(DOCX)

## Author contributions

**Conceptualization:** Seyed Mohsen Hasheminia.

**Data curation:** Bita Rasteh.

**Formal analysis:** Jafar Kolahi.

**Funding acquisition:** Seyed Mohsen Hasheminia.

**Investigation:** Masoumeh Behdarvandi.

**Methodology:** Pedram Iranmanesh.

**Project administration:** Saber Khazaei.

**Resources:** Pedram Iranmanesh.

**Software:** Jafar Kolahi.

**Supervision:** Saber Khazaei.

**Validation:** Bita Rasteh.

**Writing – original draft:** Saber Khazaei, Masoumeh Behdarvandi.

**Writing – review & editing:** Seyed Mohsen Hasheminia, Pedram Iranmanesh, Bita Rasteh.

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
