## [Decision Letter · Decision Letter 0]

Dear Dr. Behdarvandi,

Thank you for submitting your manuscript to PLOS ONE. After careful consideration, we feel that it has merit but does not fully meet PLOS ONE’s publication criteria as it currently stands. Therefore, we invite you to submit a revised version of the manuscript that addresses the points raised during the review process.

We look forward to receiving your revised manuscript.

Kind regards,

Mohammed Nasser Alhajj, BDS, MClinDent, PhD

Academic Editor

PLOS ONE

Journal Requirements:

3. Thank you for stating the following financial disclosure: “SM.H received the grant by Isfahan University of Medical Sciences (#3400125).”

4. We note that your Data Availability Statement is currently as follows: “All relevant data are within the manuscript and in Supporting Information files.”

Please confirm at this time whether or not your submission contains all raw data required to replicate the results of your study. Authors must share the “minimal data set” for their submission. PLOS defines the minimal data set to consist of the data required to replicate all study findings reported in the article, as well as related metadata and methods (https://journals.plos.org/plosone/s/data-availability#loc-minimal-data-set-definition). For example, authors should submit the following data: - The values behind the means, standard deviations and other measures reported; - The values used to build graphs; - The points extracted from images for analysis. Authors do not need to submit their entire data set if only a portion of the data was used in the reported study. If your submission does not contain these data, please either upload them as Supporting Information files or deposit them to a stable, public repository and provide us with the relevant URLs, DOIs, or accession numbers. For a list of recommended repositories, please see https://journals.plos.org/plosone/s/recommended-repositories. If there are ethical or legal restrictions on sharing a de-identified data set, please explain them in detail (e.g., data contain potentially sensitive information, data are owned by a third-party organization, etc.) and who has imposed them (e.g., an ethics committee). Please also provide contact information for a data access committee, ethics committee, or other institutional body to which data requests may be sent. If data are owned by a third party, please indicate how others may request data access.

6. Please ensure that you refer to Figure 3 in your text as, if accepted, production will need this reference to link the reader to the figure.

7. As required by our policy on Data Availability, please ensure your manuscript or supplementary information includes the following: A numbered table of all studies identified in the literature search, including those that were excluded from the analyses. For every excluded study, the table should list the reason(s) for exclusion. If any of the included studies are unpublished, include a link (URL) to the primary source or detailed information about how the content can be accessed. A table of all data extracted from the primary research sources for the systematic review and/or meta-analysis. The table must include the following information for each study: Name of data extractors and date of data extraction Confirmation that the study was eligible to be included in the review. All data extracted from each study for the reported systematic review and/or meta-analysis that would be needed to replicate your analyses. If data or supporting information were obtained from another source (e.g. correspondence with the author of the original research article), please provide the source of data and dates on which the data/information were obtained by your research group. If applicable for your analysis, a table showing the completed risk of bias and quality/certainty assessments for each study or outcome. Please ensure this is provided for each domain or parameter assessed. For example, if you used the Cochrane risk-of-bias tool for randomized trials, provide answers to each of the signalling questions for each study. If you used GRADE to assess certainty of evidence, provide judgements about each of the quality of evidence factor. This should be provided for each outcome. An explanation of how missing data were handled. This information can be included in the main text, supplementary information, or relevant data repository. Please note that providing these underlying data is a requirement for publication in this journal, and if these data are not provided your manuscript might be rejected.

Additional Editor Comments:

Dear authors

Thank you for submitting your manuscript to PLOS One journal.

The manuscript has been now reviewed by 2 independent reviewers who had some comments that should be addressed.

Kindly go through the comments and respond to all ccomments point by point.

We are looking forward to receiving the revised version.

Regards,

Reviewers' comments:

Reviewer's Responses to Questions

**Comments to the Author**

1. Is the manuscript technically sound, and do the data support the conclusions?

Reviewer #1: Partly

Reviewer #2: Yes

2. Has the statistical analysis been performed appropriately and rigorously?

Reviewer #1: N/A

Reviewer #2: Yes

3. Have the authors made all data underlying the findings in their manuscript fully available?

Reviewer #1: Yes

Reviewer #2: Yes

4. Is the manuscript presented in an intelligible fashion and written in standard English?

Reviewer #1: Yes

Reviewer #2: Yes

Reviewer #1: Thank you for giving me the chance to evaluate this work

Prevalence of Second Mesiobuccal Canal in Maxillary Molars of Iranian Population:Systematic Review with Meta-Analysis

Abstract

ok

Key words

better to be in alphabetical order

Introduction:

In general, not tight enough and need improvements, specially citing the recent and related references..

Better to go shortly highlighting what has been done in your population and how many studies and technique used in them.

then, the limitation present the current studies in your population, and clear the aim of the study

Missing the second mesiobuccal (MB2) canal was suggested as one of the most common failure

reasons for root canal treatment in maxillary molars (1).

The reference is not related at all, you should bring some reference from outcome studies or missed canals, or failure, Extra…..

Although most studies have shown the presence of the MB2 canal, there is no general agreement

about its prevalence in different populations (5).

Not related reference

Get one from gender differences, race differences and so on,

Better to get some systematic review studies reporting these variables.

The prevalence of MB2 in maxillary molars ranges from 10% to 95% (6),

This is a big statement, its better to mentions if is it worldwide or what exactly, also bring a new related reference, check Martins….

which might be related to the method of assessment and the race,

age, and gender of the study population (5).

Same note here the reference….

Materials and methods

ok

RESULTS:

In contrast, the lowest prevalence of MB2 canal was related to the direct

vision method with a prevalence of 27% (95%CI, 13_40; I2=0%), followed by the PA

radiography method with a prevalence of 15% (95%CI, -10_40; I2=91.17%)

SHOULD BE THE OTHER WAY AROUND

In contrast, the lowest prevalence was related to the direct vision method with a prevalence of 16% (95%CI, 4_27;I2=0%), followed by the PA radiography method with a prevalence of 4% (95% CI, -8_17;I2=40.52%)

SHOULD BE THE OTHER WAY AROUND

DISCUSSION:

Generally not bad but need to be improved,

Compare your pool outcome with other international studies and other sys reviews as well

You did good in explaining the variations between the pool of studies you collected

No limitations was mentioned in your study???

In the present study, the pooled prevalence of the MB2 canal in the maxillary first molar (60%)

was higher than the second molar (33%), which was similar to other studies (35).

poor reference, better to compare with other sys reviews worldwide, or mention more than one country to compare with…..

In various populations, a range of 25% to 93.5% and 34% to 50% have been reported for the first and second molars, respectively (S5 Table). This different range in the countries can be due to

various evaluation methods, racial groups, or geographical regions.

Where is the reference?

Most references in table S5 is old and well selected

Natural selection is also associated with phenotype changes (36). Large tooth size may be

maintained as a dominant natural selection (African population), while the reduction in tooth

size occurs in the absence of such force (37).

What do you mean by these sentences, better to clarify in the texts….

Further implication:

Future studies are suggested by accurately reporting the age, gender, and geographic location of the teeth, as well as using modern techniques, such as the microscope, loupe, and micro-CT.

CBCT IN VIVO IS THE BEST TO DESCRIBE ALL THE VARIABLES INCLUDING RIGHT AND LET, AGE, GENDER AND EXTEA…

Micro-CT is very precise but it can be traumatic and one of its imitation is the number of sample Versus CBCT Where you can get a very big sample

Conclusion:

The prevalence of the MB2 canal was higher in the first molar (60%) than in the second molar

(33%) of the maxilla, and the prevalence of the MB2 canal depended on the assessment method.

According to your study and limitations, the technique used is the main finding which make the result of the study not convincing compared to worldwide agreements of MB2

It can be improved and focused on the type of technique used

Reviewer #2: A good study has been done and most of the subjects that should be done in systematic review studies such as publication bias, risk of bias, heterogeneity, subgroup analysis, etc. have been done. But in order to increase the quality of the article, the following items are suggested to be reviewed and corrected:

1- PRISMA 2020 framework should be used in the study.

2- Due to the fact that the resolution of the graphs is low, it is suggested to save the images of the graphs with the tif extension.

3- It is better to perform sensitivity analysis for studies with a small sample size, such as Naseri's 2015 study (sample of 35 people). In such a way that these studies are removed and recalculate the pool effect.

4- The tool used to check the risk of bias seems to be the tool for evaluating the quality of studies, rather than risk. It is better to mention this and add the score of each study in table 1 and 2 (bibliographic table).

5- The bibliographic information table is incomplete; it is better to mention the type of study and the quality score of the study.

6- Several graphs and parameters have been used for some topics, such as heterogeneity. It is better to report only the Galbraith graph for this topic.

**Do you want your identity to be public for this peer review?** For information about this choice, including consent withdrawal, please see our Privacy Policy

Reviewer #1: No

Reviewer #2: No

---

## [Author Response · Author response to Decision Letter 1]

5 Feb 2025

The reviewer comments were addressed in a table and uploaded as a file named "Response to Reviewers."

---

## [Decision Letter · Decision Letter 1]

Prevalence of Second Mesiobuccal Canal in Maxillary Molars of Iranian Population: A Systematic Review with Meta-Analysis

PONE-D-24-24127R1

Dear Dr. Behdarvandi,

We’re pleased to inform you that your manuscript has been judged scientifically suitable for publication and will be formally accepted for publication once it meets all outstanding technical requirements.

Kind regards,

Mohmed Isaqali Karobari, BDS, MScD.Endo, Ph.D. Endo, FDS, FPFA, FICD, MFDS

Academic Editor

PLOS ONE

Additional Editor Comments (optional):

Dear Authors,

The authors have addressed all the comments and suggestions reviewers gave, and the manuscript has dramatically improved. In PRISMA, the reasons for excluding articles should be clearly stated, rather than saying reason 1, 2. The manuscript can be accepted for publication in its current form. I would like to congratulate the authors and wish them all the very best in their future endeavours.

Best regards and keep well.

Reviewers' comments:

Reviewer's Responses to Questions

**Comments to the Author**

Reviewer #2: All comments have been addressed

2. Is the manuscript technically sound, and do the data support the conclusions?

Reviewer #2: Yes

3. Has the statistical analysis been performed appropriately and rigorously?

Reviewer #2: Yes

4. Have the authors made all data underlying the findings in their manuscript fully available?

Reviewer #2: Yes

5. Is the manuscript presented in an intelligible fashion and written in standard English?

Reviewer #2: Yes

Reviewer #2: In PRISMA, the reasons for excluding articles should be clearly stated, rather than saying reason 1, 2.

**Do you want your identity to be public for this peer review?** For information about this choice, including consent withdrawal, please see our Privacy Policy

Reviewer #2: **Yes: ** Dr Reza Goudarzi

---

## [Editor Report · Acceptance letter]

PONE-D-24-24127R1

PLOS ONE

Dear Dr. Behdarvandi,

I'm pleased to inform you that your manuscript has been deemed suitable for publication in PLOS ONE. Congratulations! Your manuscript is now being handed over to our production team.

Kind regards,

on behalf of

Prof Dr. Mohmed Isaqali Karobari

Academic Editor

PLOS ONE